

# Antimicrobial and micronutrient interventions for the management of infants under 6 months of age identified with severe malnutrition: a literature review

Timothy J. Campion-Smith[1], Marko Kerac[2], Marie McGrath[3] and James A. Berkley[4,5,6]

[1] Department of Paediatrics, John Radcliffe Hospital, Oxford, United Kingdom
[2] Department of Population Health, London School of Hygiene & Tropical Medicine, University of London, London, United Kingdom
[3] Emergency Nutrition Network, Oxford, United Kingdom
[4] KEMRI-Wellcome Trust Research Programme, Kilifi, Kenya
[5] The Childhood Acute Illness & Nutrition (CHAIN) Network, Nairobi, Kenya
[6] Centre for Tropical Medicine and Global Health, University of Oxford, Oxford, United Kingdom

Corresponding author
Timothy J. Campion-Smith, tcampionsmith@doctors.org.uk

## ABSTRACT

**Background.** Infants under 6 months (U6M) contribute a significant proportion of the burden and mortality of severe malnutrition globally. Evidence of underlying aetiology in this population is sparse, but it is known that the group includes ex-preterm and low birthweight (LBW) infants. They represent a unique population given their dependence on breastmilk or a safe, secure alternative. Nutrition agencies and health providers struggle to make programming decisions on which interventions should be provided to this group based upon the 2013 WHO Guidelines for the 'Management of Severe Acute Malnutrition in Infants and Young Children' since there are no published interventional trial data focussed on this population. Interim guidance for this group might be informed by evidence of safety and efficacy in adjacent population groups.

**Methodology.** A narrative literature review was performed of systematic reviews, meta-analyses and randomised controlled trials of antimicrobial and micronutrient interventions (antibiotics, deworming, vitamin A, vitamin D, iron, zinc, folic acid and oral rehydration solution (ORS) for malnutrition) across the population groups of low birthweight/preterm infants, infants under 6 months, infants and children over 6 months with acute malnutrition or through supplementation to breastfeeding mothers. Outcomes of interest were safety and efficacy, in terms of mortality and morbidity.

**Results.** Ninety-four articles were identified for inclusion within this review. None of these studied interventions exclusively in severely malnourished infants U6M. 64% reported on the safety of studied interventions. Significant heterogeneity was identified in definitions of study populations, interventions provided, and outcomes studied. The evidence for efficacy and safety across population groups is reviewed and presented for the interventions listed.

**Conclusions.** The direct evidence base for medical interventions for severely malnourished infants U6M is sparse. Our review identifies a specific need for accurate micronutrient profiling and interventional studies of micronutrients and oral fluid management of diarrhoea amongst infants U6M meeting anthropometric criteria

for severe malnutrition. Indirect evidence presented in this review may help shape interim policy and programming decisions as well as the future research agenda for the management of infants U6M identified as malnourished.

## INTRODUCTION

Malnutrition remains a major problem globally with an estimated 22.4 million children under 5 years old affected by severe wasting (*Institute for Health Metrics and Evaluation, 2015*), and with all forms of undernutrition being a causal factor in 45% of childhood mortality (*Black et al., 2013*). Recently, there has been increasing awareness that infants under 6 months (U6M) make up a sizable proportion of these children, with 3.8 million infants U6M estimated as severely wasted (weight-for-length <3 standard deviations below median, as per current WHO definition) and 4.7 million moderately wasted (weight-for-length between three and two standard deviations below median) (*Kerac et al., 2011*). Mortality rates are higher in this group compared to their counterparts aged over 6 months (*Grijalva-Eternod et al., 2017*). However, the total number of infants U6M with any anthropometric indicator of malnutrition (including moderate wasting, stunting, low weight-for-age or low mid upper arm circumference (MUAC)) and who are therefore at increased risk of morbidity and mortality, is significantly greater. This population group typically includes some infants who were born preterm or with low birthweight (LBW) as well as those with primary or secondary (i.e., resulting from illness) malnutrition (*Mwangome et al., 2017*; *Mwangome et al., 2019*). This is a unique population who are nutritionally dependent on breastmilk (or a safe, secure alternative), whilst at increased risk of serious common infections compared to older children as their immunity develops. Hence, they do not fit neatly into either neonatal nutritional guidelines or guidelines for therapeutic feeding programmes among children over 6 months of age.

The 2013 update of the World Health Organization (WHO) guidelines on Severe Acute Malnutrition (SAM) was the first to include specific recommendations for infants U6M, accommodating both inpatient and outpatient care of complicated (i.e., clinically unstable as well as malnourished) and uncomplicated (i.e., clinically still stable despite anthropometric compromise) respectively (*World Health Organization, 2013a*). These guidelines focus on acute malnutrition as defined by low weight-for-length, however there is increasing recognition that other anthropometric indicators (low weight-for-age, low MUAC, mid upper arm circumference) may better identify high-risk infants (*Lelijveld et al., 2017*). Hence, in this paper, we deliberately use the term 'severe malnutrition' to identify our target group: those small infants U6M who, as per WHO 2013 SAM guidelines, are at high risk of mortality, morbidity and future neurodevelopmental impairment (*Bhutta et al., 2017*).

Whilst the WHO 2013 guidelines for managing malnutrition among infants U6M make numerous recommendations, the key focus is on feeding support, with the aim of re-establishing effective exclusive breastfeeding. Other important aspects of care are less well described, with suggestions that infants U6M receive *"the same general medical care as infants with severe acute malnutrition (SAM) who are 6 months of age or older* (*World Health Organization, 2013a*). Thus, there is ambiguity in how these guidelines should be interpreted and implemented, making programming difficult for nutrition agencies, policymakers and health providers. This review was directly prompted by questions to the Emergency Nutrition Network (ENN) from nutrition agencies and health providers as to which interventions besides breastfeeding support are safe, effective and should be prioritised (e.g., antibiotic treatment, deworming, micronutrient supplementation and oral rehydration) in the management of acutely malnourished infants U6M (*NNE, 2018*).

Recommendations on the management of severe malnutrition in infants U6M are currently hampered by lack of evidence in this age group, acknowledged in the WHO 2013 SAM guideline update (*World Health Organization, 2013a*). Besides shorter-term risks of death and morbidity, there are important long-term consequences for neurodevelopment and adult non-communicable disease (*Tarry-Adkins & Ozanne, 2011*). The determinants of severe malnutrition in infants U6M are only just beginning to be studied (*Munirul Islam et al., 2018*). It is not yet clear what proportion of infants are presenting with acute wasting as opposed to following a trajectory from LBW, whether premature or small for gestational age (SGA), and whether this has implications for assessment and treatment.

In the absence of specific evidence, interim guidance might be informed by the evidence and guidelines developed for infants close in age (LBW) or nutritional status (malnourished children age 6 months or more). By way of background, the current WHO recommendations for micronutrient supplementation in very LBW (VLBW)/LBW infants and for children with SAM are summarised in Table 1.

This literature review aims to summarise evidence for antimicrobial and micronutrient treatments that might inform policies for management of infants U6M identified as severely malnourished. Objectives towards this are: to identify evidence from adjacent paediatric populations (LBW infants and infants and children aged >6 months with severe malnutrition) on micronutrient supplementation and antimicrobial interventions; to summarise what this evidence shows in terms of safety and efficacy; and to identify critical areas needing future research.

Key abbreviations and definitions used throughout the article are presented in Fig. 1. Whilst our focus is on infants U6M with severe malnutrition, where an article is cited within a malnourished population, that article's terminology with regard to malnutrition is presented in inverted commas. For ease of interpretation, we present data in the following age categories, LBW/preterm, infants U6M, and children over 6 months (up to 18 years but with the majority of included articles focusing on 6-59months). This is to reflect the structure the current WHO SAM guidelines as U6M and over 6 months, recognising that a significant proportion of the at-risk U6M population may have been born LBW/preterm or both.

**Table 1  Summary of current recommendations for antimicrobial and micronutrient supplementation amongst LBW/VLBW infants and children with SAM (WHO) (*World Health Organization, 2013a*; *World Health Organization, 2011*).**

| Intervention | VLBW/LBW infants | Children (aged 6–59 months) with SAM |
| --- | --- | --- |
| **Antibiotics** | – | Yes |
| **Deworming** | – | Antihelminthics during rehabilitation phase if high prevalence region or evidence of infestation |
| **Vitamin A** | Not recommended | 5,000 IU daily<br>High dose regimen (50,000–200,000 IU) if eye signs or recent measles |
| **Vitamin D** | 400–1000 IU/day until 6 months of age[*] | No recommendation<br>Therapeutic feeds provide 135-300 IU/kg/day[**] |
| **Folic Acid** | – | 5 mg day 1, 1mg daily thereafter[+] |
| **Iron** | 2–4 mg/kg/day from 2 weeks to 6 months of age[*] | 3 mg/kg after 2 days on F-100 formula. (Not if receiving RUTF) |
| **Zinc** | Not recommended | 2 mg/kg/day[+] |
| **Copper** | – | 0.3 mg/kg/day[+] |
| **Calcium** | 120–140 mg/kg/day if breastmilk fed[*] | No recommendation<br>Therapeutic feeds provide:100 mg/kg/day[**] |
| **Phosphorous** | 60–90 mg/kg/day day if breastmilk fed[*] | No recommendation<br>Therapeutic feeds provide: 100 mg/kg/day[**] |

Notes.

VLBW, very low birth weight; SAM, severe acute malnutrition.

[*]Recommendation specifically for VLBW infants only.

[**]Range based upon composition of commercial $F-75/F-100$ at 130 ml/kg/day volumes.

[+]If not on therapeutic milk (e.g., F75/F100) or ready-to-use therapeutic food (RUTF).

## Survey methodology

In this scoping review, PubMed and Google Scholar databases were searched on 1st October 2018 to identify systematic reviews (SRs), meta-analyses (MAs) and randomised-controlled trials (RCTs) where the full-text was available in English. Populations included in the search criteria were infants U6M (with or without severe malnutrition), infants and children over the age of 6 months (with or without severe malnutrition), pre-term/LBW/VLBW infants and breastfeeding mothers. Any definition of severe malnutrition was included. The interventions included were those mentioned in the current WHO guidelines for the management of SAM in those over the age of 6 months (*World Health Organization, 2013a*) and for the management of LBW/VLBW infants (*World Health Organization, 2011*): antibiotics, deworming, vitamin A, vitamin D, iron, zinc, folic acid and oral rehydration solution (ORS) for malnutrition (ReSoMal), as well as maternal micronutrient and macronutrient supplementation in the post-natal period up to 6 months of age (not from WHO guideline). Outcomes of interest were mortality, morbidity, anthropometric changes, neurodevelopment and adverse effects on the infant or child.

The following were considered outside the scope of this review: breast milk fortifiers; pre-natal nutritional supplementation; breastfeeding support interventions; and the duration, mechanism of delivery and cost of interventions.

Identified articles were screened for suitability and reviewed by the first author. Where the results of an RCT were included and presented in a subsequent MA, the article was not included unless it contained a subgroup population of interest (e.g., malnourished

**Child –** Those 12 months or older (upper age limit not defined)

**EBF** – Exclusive breastfeeding

**Infant(s)** – Children under 12 months of age. Where infants under 6 months of age are referenced, this is specified as U6M.

**Low birthweight (LBW)** - Birth weight less than 2.5kg. LBW can be a consequence of preterm birth, or due to small for gestational age (SGA), or both[13]

**IPTp** – intermittent preventive treatment in pregnancy (malaria)

**IV** - Intravenous

**MA** – Meta-analysis

**MAM** – Moderate acute malnutrition*

**MUAC** – mid-upper-arm circumference

**ORS** – oral rehydration salts/solution

**Pre-term** – Birth before 37 completed weeks of gestation[12]

**ReSoMal** – oral rehydration solution for severely malnourished children

**RCT** - Randomised-controlled trial

**RUTF** – Ready-to-use therapeutic food

**SAM** – Severe acute malnutrition*

**Small for gestational age (SGA)** - Weight for gestation <$10^{th}$ percentile [13]

**SMC** – Seasonal malarial chemoprophylaxis

**SR –** Systematic review

**U6M** – Under six months of age

**WAZ** – Weight-for-age z-score

**WHZ/WLZ –** weight-for-height/-length z-score

**Very Low Birthweight (VLBW)** - birthweight less than 1.5kg[13]

**Figure 1  Abbreviations and definitions.** * Standardised definitions of malnutrition have not been specified by the authors for this review and there is variation in the metrics included studies have used to define these.

infants and children) that was not presented or the primary population of interest in the MA. Similarly, where a study looked primarily at multiple micronutrient interventions, this was included if the effects of including or excluding a single micronutrient. When LBW is mentioned, it refers to studies that defined a population as either SGA, pre-term or LBW (<2.5 kg). When VLBW is mentioned, it refers specifically to a study of <1.5 kg birth weight.

A formal GRADE assessment for quality of evidence of all included articles was beyond the scope of this review but where a GRADE classification of quality (Very Low/Low/Moderate/High) has been made in a recent SR or MA, this is presented.

## RESULTS

We identified a total of 94 articles for inclusion and review. None of these focused exclusively on severely malnourished infants U6M. Four articles (4%) examined infants U6M alongside other malnourished children over 6 months; 15 (16%) RCTs included malnourished children (both U6M and up to age 14 years). No subgroup analysis by age was presented in any of these articles. Figure 2 displays the distribution of evidence by population of interest and by study type.

Of included articles, 60 (64%) documented the presence or absence of adverse effects or addressed safety. Table 2 presents proportion of articles reporting on safety, by intervention.

Our search identified marked heterogeneity in: age range of the population studied; anthropometric definitions of malnutrition; dosage and duration of intervention; and outcomes studied. This makes summarising overall direction and size of effect challenging. Key findings by intervention are shown in Table 3.

### Antibiotics

Current WHO guidance for antibiotic use in management of 'SAM' in infants U6M is to treat inpatient admissions (i.e., those with 'complicated' disease) with intravenous (IV) antibiotics and outpatients (i.e., those with 'uncomplicated' disease) with oral antibiotics, such as amoxicillin. Our search identified eight key articles for inclusion in this review (five in malnourished children over 6 months (two SRs, one MA and three RCTs), one in LBW infants and one published RCT that included malnourished infants U6M). Summarising a literature-base that includes multiple drug classes with different pharmacokinetics and potential side effects is challenging but is addressed in two SRs on this topic in malnourished children (*Lazzerini & Tickell, 2011*; *Williams & Berkley, 2018*).

Arguments framing the usage of antibiotics for malnourished infants U6M include the higher mortality rates in this population (*ENN/UCL/ACF, 2010*) and the higher prevalence of community-acquired bacteraemia in infants compared to older children amongst paediatric hospital admissions (*Berkley et al., 2005*). Also important are the costs of antibiotic use, both short-term financial costs, but also longer-term family and societal costs, with increasing use contributing to antimicrobial resistance (*Ferri et al., 2017*).

### *Efficacy*

In the management of 'SAM' in children over 6 months, a recent MA and comment on two RCTs assessing the efficacy of oral amoxicillin compared to placebo reported a 3% increase in survival with antibiotic treatment with amoxicillin (*Million, Lagier & Raoult, 2017*), whilst the individual trials noted an increased rate of MUAC gain with amoxicillin (*Trehan et al., 2013*), improved rate of recovery and decreased risk of transfer to inpatient care (*Isanaka et al., 2016*). An older RCT assessed the efficacy of two days of intramuscular ceftriaxone in comparison to five days of oral amoxicillin and suggested benefits in terms of growth, recovery rate and case fatality rate in the ceftriaxone treated group, but this was not statistically significant (*Dubray et al., 2008*).

The only study to include infants U6M (as well as those over 6 months) with 'SAM' examined 12 month mortality with or without six months of co-trimoxazole prophylaxis

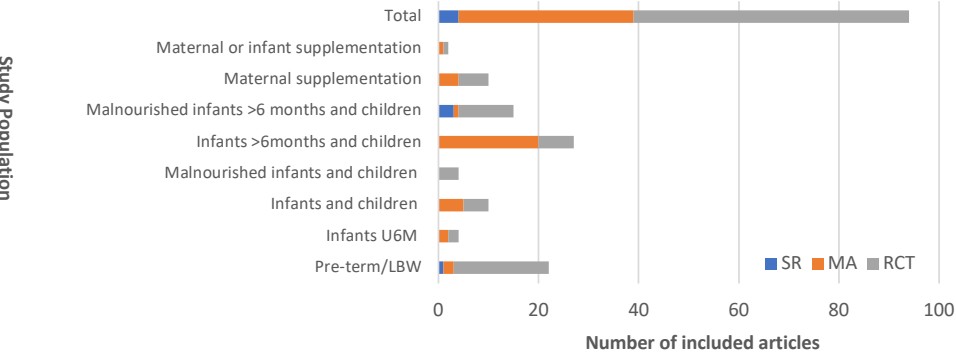

**Figure 2** **Distribution of included articles presented by population studied and study design.** SR, Systematic Review; MA, Meta-analysis; RCT, randomised controlled trial. 'Malnourished' describes any definition of severe malnutrition.

**Table 2** **Distribution of studies reporting on adverse effects by antimicrobial or micronutrient intervention.**

| Intervention | Adverse effects documented | Adverse events not documented | Total |
|---|---|---|---|
| **Antibiotics** | 8 | 0 | 8 |
| **Deworming** | 4 | 4 | 8 |
| **Vitamin A** | 6 | 1 | 7 |
| **Vitamin D** | 13 | 3 | 16 |
| **Iron** | 8 | 5 | 13 |
| **Zinc** | 10 | 14 | 24 |
| **Folic Acid** | 6 | 4 | 10 |
| **Maternal supplementation** | 2 | 3 | 5 |
| **ReSoMal** | 3 | 0 | 3 |
| **Total** | 60 | 34 | 94 |

after inpatient admission for 'SAM'. This found no evidence of mortality or growth benefits (*Berkley et al., 2016*).

With regard to inpatient IV therapy for 'complicated SAM', Williams & Berkley identified one interventional trial from 1996 and argue that current recommendations are based on no supporting evidence for either the type and duration of antibiotics (*Williams & Berkley, 2018*).

In the pre-term population, one study provides evidence of a role for erythromycin in establishing exclusive breastfeeding (EBF) in pre-term infants (*Gokmen et al., 2011*), likely secondary to its gastrointestinal prokinetic effects (a potential avenue for a future package of interventions in this group). Indeed, there is increasing interest in the potential roles of macrolides in reducing childhood mortality. A cluster-randomised trial based in Tanzania, Malawi and Niger reported in 2018 an overall reduction of mortality by 13.5% in those communities receiving twice-yearly mass administration of azithromycin compared to those receiving placebo. Of particular note is that the largest reductions in mortality were

Peer J

**Table 3  Summary of results by antimicrobial and micronutrient intervention.**

**Antibiotics:**

Current evidence, high mortality rates, higher rates of bacteraemia in the malnourished infant population and lower specificity of clinical signs for serious infections than in older age groups make divergence from current guidelines difficult to justify for infants U6M.

Urgent research is required on this topic, especially for those infants who appear clinically stable and for whom risks and costs of routine antibiotic use may outweigh potential benefits. The potential roles for macrolide antibiotics in vulnerable populations, particularly in the context of global increases in antimicrobial resistance, also require further evaluation.

**Deworming:**

There is no evidence to support introduction of routine deworming in infants U6M based on current evidence.

There is some evidence for deworming in breastfeeding mothers of malnourished infants U6M that requires further evaluation.

**Vitamin A:**

Low-dose supplementation shows the potential for significant benefit in terms of mortality and diarrhoea incidence in deficient populations and in such settings should be given. However, outside of such situations of specific clinical need, routine use cannot be currently recommended given strong evidence of mild to moderate side-effects.

More research on which populations/individuals do and which do not need extra vitamin A would be valuable.

**Vitamin D:**

Supplementation is safe at doses reviewed within this report with evidence of efficacy in terms of growth in children over 6 months with 'uncomplicated SAM' and LBW infants, reduced morbidity in children and through maternal supplementation, potential roles in sustaining EBF. Given the fact that a considerable proportion of infants U6M were born at LBW (*Kerac et al., 2019*), the current WHO recommendations for LBW of 6 months supplementation can reasonably be followed in nutritional programming for malnourished infants U6M where birth weight is unknown

Further trials of vitamin D in malnourished infants U6M who are not LBW are warranted. There are also questions about the optimum dose, duration and mode of delivery.

**Iron:**

There is a lack of any strong evidence for benefits of iron supplementation in terms of mortality and morbidity but evidence of increased haemoglobin status and some neurodevelopmental benefits across age groups. Concerns, such as those raised in the WHO guidelines on iron supplementation in children (*Sachdev, Gera & Nestel, 2005*), exist about potential negative impacts of iron supplementation on growth, infection risk and malaria risk in malaria-endemic settings where regions where malaria prevention and treatment systems are not in place and/or where children are already iron replete (*Lönnerdal, 2017*). Routine use for all malnourished infants U6M cannot therefore be recommended, but there can be exceptions for specific individuals and/or populations for treatment of iron deficiency.

Further trials are investigating potential alternatives to simple iron salts and ways to target iron therapy.

**Zinc:**

Consistent evidence across age groups exists of zinc supplementation being associated with reduced morbidity and improved anthropometry, whilst mortality and neurodevelopmental impacts are more unclear. Zinc should not be supplemented as a high-dose, given mortality concerns, and its tolerability should be considered in the context of vomiting risk. Zinc should be supplemented as per diarrhoea guidelines for all severely malnourished infants U6M with diarrhoeal illnesses (*World Health Organization, 2005*). In cases where severely malnourished infants U6M are not affected by diarrhoea, we suggest that 2mg/day (1 recommended daily allowance (*Institute of Medicine , US*)) of zinc be supplemented with other micronutrients in regions where zinc deficiency has been documented. This is because breastmilk zinc concentrations have been shown to be insufficient in zinc deficient mothers (*Dumrongwongsiri et al., 2015*; *Yalcin, Yalcin & Gucus, 2015*).

Research is urgently needed to establish zinc requirements for malnourished infants U6M.
**Table 3** (*continued*)

**Folic acid:**

Limited evidence of benefit in the child population in terms of morbidity and growth and therefore this is not recommended as a routine intervention for malnourished infants U6M. Safety in a malaria endemic setting remains uncertain but is less concerning given SP is no longer routinely used for the treatment of malaria. Safety of folic acid supplementation should be considered in areas where SP is used for Seasonal Malarial Chemoprophylaxis (SMC) or Intermittent Preventive Treatment in pregnancy (IPTp).

Further studies investigating the role of folic acid in malnourished infants U6M are warranted but the evidence presented does not identify this as a priority area for research.

**Maternal macro- and micro-supplementation:**

We found insufficient evidence of the benefits of maternal supplementation to infants to justify routine use in current programming for malnourished infants U6M.

Potential benefits to the mother, not included in this review, should be considered and evaluated in more detail in further research to inform decisions in this area.

**ReSoMal/ORS:**

There are no studies of ReSoMal in infants U6M. Limited evidence from inpatient studies of older children with 'SAM' suggests that ReSoMal is of similar efficacy in terms of rehydration to standard ORS but that there are significant safety concerns in terms of risk of hyponatraemia. On the basis of current evidence, and the fact that infants in the first few months of life are at increased risk of water and salt retention due to immature hormonal and renal excretion mechanisms, there is no reason to change current recommendations for use of ReSoMal in malnourished infants U6M.

This age group may differ from older children in both risks and responses to treatment and is thus a priority area for clinical trials.

seen in the infants 1-5 months group (nutritional status not reported) and in the most disadvantaged communities (*Keenan et al., 2018*).

### Safety

No study addressed the safety profile of oral amoxicillin or first-line parental antibiotics in malnourished infants U6M. Co-trimoxazole prophylaxis was associated with an increased risk of neutropenia (grade 4), but age-specific effects were not reported (*Berkley et al., 2016*).

Williams & Berkley's systematic review addressed the safety profiles of antibiotics used in the included studies, showing there was not a *"significant rate of adverse effects documented in antibiotic intervention group(s)"*, but they do raise safety concerns about the currently recommended seven day course of gentamicin in the 'SAM' population, given its known renal toxicity, uncertain pharmacokinetics in the 'SAM' population and inability to monitor renal function and serum concentrations in many low-income settings (*Williams & Berkley, 2018*).

### Summary

Current evidence, high mortality rates, higher rates of bacteraemia in infants than older children and lower specificity of clinical signs for serious infections than in older age groups make divergence from current guidelines difficult to justify for infants U6M.

Urgent research is required on this topic, especially for those infants who appear clinically stable and for whom risks and costs of routine antibiotic use may outweigh potential benefits. As one SR comments *"given that these antibiotics have side-effects, costs, and risks as well as benefits, their routine use needs urgent testing…there is sufficient equipoise for placebo controlled RCTs."* (*Alcoba et al., 2013*) The potential roles for macrolide antibiotics in vulnerable populations, particularly in the context of global increases in antimicrobial resistance, also require further evaluation. Mass antibiotic distribution using azithromycin may become policy in some low resource settings following recent trial and risks relating to antimicrobial resistance will need careful monitoring.

## Deworming

Given limited mobility and the recommendation for EBF up to 6 months of age, it is often assumed that helminth burden is acquired later in infancy. However, evidence suggests that helminth acquisition may occur in infants U6M (*Ghiwot, Degarege & Erko, 2014*; *Fonseca et al., 2014*; *Goto, Mascie-Taylor & Lunn, 2009*). Whether or how this affects severe malnutrition at this age has not been investigated. Furthermore, anti-helminthic agents are not typically licensed for use in infants U6M.

We identified four MAs of relevance in the child population, three controlled trials in children with severe malnutrition (one which includes a sample of infants U6M) and one RCT that studied deworming in breastfeeding mothers.

### Efficacy

In children over 6 months, the benefits of mass deworming are debatable, with the included MAs providing evidence that it offers no benefit in terms of mortality, nutritional status

or cognition (GRADE Low/Moderate) (*Welch et al., 2017*; *Taylor-Robinson et al., 2015*; *Clarke et al., 2017*; *Thayer, Clermont & Walker, 2017*), although based on trials carried out in heterogenous populations with differing disease prevalence, age profiles and with nutritional status. Among individual studies, in malnourished pre-school aged children, one cluster-randomised trial in India showed significant improvements in weight gain amongst stunted and wasted infants treated with albendazole ($p < 0.0001$) (*Awasthi et al., 2008*), whereas an RCT amongst 222 children under 5 years in the Democratic Republic of Congo showed negative impacts of mebendazole treatment on weight, height and MUAC gain ($p$-values 0.002, 0.028 and 0.012 respectively) (*Donnen et al., 1998a*). More recently, in Malawi, a cluster-randomised trial of a package of interventions, that included deworming as a component, at discharge from supplementary feeding programmes for 'moderate acute malnutrition (MAM)' showed no significant impact on rates of 'MAM' relapse at 12-month follow-up (*Stobaugh et al., 2017*).

Relevant to malnourished infants U6M, we identified one RCT that looked at infant outcomes with albendazole treatment of breastfeeding mothers, showing a substantial mean difference gain of 0.5 (95% CI [0.2–0.8], $p = 0.003$) in length-for-age Z score (LAZ) in infants of treated mothers who had stool smears positive for helminth infections (*Mofid et al., 2017*).

### Safety
Deworming treatment was associated with no adverse effects in included studies. WHO recommend deworming from the age of 12 months (*World Health Organization, 2019*). This is because of the presumed epidemiology of helminth infections and although there is less safety data there are no specific additional safety concerns in younger infants (*Montresor, Awasthi & Crompton, 2003*).

### Summary
There is no evidence to support introduction of routine deworming in infants U6M based on current evidence.

There is some evidence for deworming in breastfeeding mothers of malnourished infants U6M that requires further evaluation.

## Vitamin A
Vitamin A deficiency is a known cause of xerophthalmia and blindness and is associated with increased mortality from diarrhoeal disease and measles. The most recent estimate of the prevalence of vitamin A deficiency amongst children aged 6-59 months in low- and middle-income countries is 29%, ranging from 48% in sub-Saharan Africa to 6% in East & Southeast Asia and Oceania, and contributes to 1.7% of all-cause mortality (*Stevens et al., 2015*). High quality epidemiological studies focusing on vitamin A profiles amongst infants or children with severe malnutrition are lacking. Breastfeeding has been shown to be a protective factor for xeropthalmia (*Semba et al., 2004*).

We identified five MAs across all populations and two RCTs recruiting only children with 'SAM'. There was significant variation in dose and dosing regimen. 'High dose' ranged from 50,000-200,000IU in children over 12 months, from 25,000-100,000IU in

infants under 12 months and from 2000-10,000IU in LBW. The most frequent 'low dose' was 5000IU in infants and children.

### Efficacy

In the U6M population, neither MA identified any impact of vitamin A on mortality (GRADE - Moderate/High) (*Imdad, Ahmed & Bhutta, 2016*; *Haider, Sharma & Bhutta, 2017*). No impact was noted on diarrhoea point prevalence in one of these (GRADE –Moderate) (*Imdad, Ahmed & Bhutta, 2016*). In the child population over the age of 6 months, one MA reported there to be strong, evidence of vitamin A supplementation being associated with reduction in all cause- and diarrhoea-associated mortality by 12% (GRADE –High) and of diarrhoea incidence by 15% (Grade –Low) (*Imdad et al., 2017*).

In VLBW infants, a MA shows a similar level of reduction in all-cause mortality but this is not statistically significant (GRADE –Moderate) (*Darlow, Graham & Rojas-Reyes, 2016*). MA evidence of maternal post-partum vitamin A supplementation reported no benefit in terms of infant morbidity or mortality outcomes (GRADE - Very Low/Low) (*Oliveira, Allert & East, 2016*).

Among children 0-72 months of age with 'MAM', one RCT of single high dose of vitamin A compared to placebo (*Donnen et al., 1998b*) reported increases in annual weight gain (Mean Difference 0.91 kg, $p = 0.029$) and in annual MUAC gain (Mean Difference 1.29 cm, $p = 0.012$) with but there seems to be no benefit of high dose supplementation compared to low dose in other RCTs (*Donnen et al., 1998b*; *Sattar et al., 2012*; *Donnen et al., 2007*).

### Safety

Adverse effects were noted in both infant and child populations. Vitamin A supplementation was shown to be associated with a 1.5–3 times increased risk bulging of the anterior fontanelle 48–72 h after first dosing. In all cases this resolved spontaneously and was associated with no neurological sequelae (GRADE –High) (*Imdad, Ahmed & Bhutta, 2016*; *Haider, Sharma & Bhutta, 2017*). A two times increased risk of one or more episode of vomiting on commencing supplementation was noted in infants and child between the ages of 6-72 months (GRADE –Moderate) (*Imdad et al., 2017*), and on subgroup analysis in two RCTs there was some evidence of an elevated risk of diarrhoea in children of 6-60 months age without 'SAM' (*Fawzi et al., 2000*) or with high-dose vitamin A in 'malnourished pre-school children without oedema' (*Donnen et al., 1998b*).

### Summary

Low-dose supplementation shows the potential for significant benefit in terms of mortality and diarrhoea incidence in deficient populations and in such settings should be given. However, outside of such situations of specific clinical need, routine use cannot be currently recommended given strong evidence of mild to moderate safety concerns.

More research on which populations/individuals do and which do not need extra vitamin A would be valuable.

## Vitamin D

Vitamin D has diverse effects, being involved in calcium homeostasis, immune modulation, cell metabolism and growth. Global worldwide estimates of vitamin D deficiency in unsupplemented breastfed infants is 76% (18–82%) (*Dawodu & Wagner, 2012*).

Our search identified three MAs across infants, children and mothers and 13 RCTs across all populations. Six (one MA and five RCTs) of the included studies focused on serum vitamin D sufficiency as the outcome of interest. These were included as evidence for safety (*Winzenberg et al., 2011*; *Tergestina et al., 2016*; *Hollis et al., 2015*; *Oberhelman et al., 2013*; *Wheeler et al., 2016*; *Huynh et al., 2017b*). Doses of vitamin D were heterogeneous across studies, varying from 200-1000IU in LBW, between 400IU daily and a 50,000IU bolus in infants U6M, between 402IU daily and 200,000IU as a single bolus in children, and from 1200-5000IU in breastfeeding mothers.

### *Efficacy*

The only RCT, conducted in the urban setting in India, to supplement breastfed infants U6M, either directly orally or by supplementing breastfeeding mothers orally, showed no significant difference in weight, length or head circumference between vitamin D and placebo. However, the mean number of days with respiratory or diarrhoeal illness was reduced by 33.5 days by 9 months of life ($p < 0.05$) in infants supplemented with vitamin D orally (*Chandy et al., 2016*).

In children over 6 months, one recent RCT carried out in 'uncomplicated SAM' in Pakistan was identified. Children received either 2 high doses of vitamin D or placebo with follow-up at 8 weeks. This study showed significant anthropometric effects in those receiving vitamin D, with increases in weight-for height z score (WHZ)/weight-for-length z score (WLZ) (adjusted mean difference: 1.07; 95% CI: 0.49,1.65, $P < 0.001$), and significant improvements across multiple developmental indices (gross motor, fine motor and language) (*Saleem et al., 2018*). In MAs of children under 5 years, no impact was noted on growth metrics, incidence of pneumonia (GRADE –Moderate), recovery time from pneumonia (GRADE –Low) and pneumonia-specific (GRADE –Very Low) and all-cause mortality (GRADE –Low) rates (*Yakoob et al., 2016*; *Rooze et al., 2016*; *Das, Singh & Naik, 2018*).

In VLBW/LBW infants, vitamin D supplementation resulted in increases in all growth metrics (height, weight, MUAC) in two RCTs (*Kumar et al., 2011*; *Mathur, Saini & Mishra, 2016*) but follow-up of one of these studies at 3–6 years showed no lasting difference between groups (*Trilok-Kumar et al., 2015*) and one RCT in extremely pre-term infants (23-27 weeks gestation) showed no impact in terms of mortality (*Fort et al., 2016*).

Two RCTs of maternal vitamin D supplementation showed 13% increased prevalence of reported EBF at 6 months, a mean of 28 fewer days of respiratory and diarrhoeal illness by age 9 months ($p < 0.01$), but no impact on infant growth (*Chandy et al., 2016*; *Czech-Kowalska et al., 2014*).

### *Safety*

No adverse effects noted with vitamin D supplementation in any of the included studies.

*Summary*

Vitamin D supplementation is safe at doses reviewed within this report with evidence of efficacy in terms of growth in children over 6 months with 'uncomplicated SAM' and LBW infants, reduced morbidity in children and through maternal supplementation, potential roles in sustaining EBF. Given the fact that a considerable proportion of infants U6M were born at LBW (*Kerac et al., 2019*), the current WHO recommendations for LBW of 6 months supplementation can reasonably be followed in nutritional programming for malnourished infants U6M where birth weight is unknown.

Further trials of vitamin D in malnourished infants U6M who are not LBW are warranted. There are also questions about the optimum dose, duration and mode of delivery.

## Iron

Iron has been implicated not just in haemoglobin synthesis but also in muscular, neurological and immune development (*Beard, 2001*). However, debate continues around whether iron deficiency is protective of malaria and conversely whether iron excess increases risk of severe malaria (*Neuberger et al., 2016*). Further concerns exist as to where iron supplementation is implicated in increased risk of bacterial infections in newborns (*Brabin, Brabin & Gies, 2013*), risk of diarrhoea in infants and children, a change in gut flora and increased gastrointestinal inflammation and subsequent morbidity (*Paganini, Uyoga & Zimmermann, 2016*).

The global burden of anaemia in children and infants is estimated to be 41.8% (*Mclean et al., 2008*) and amongst the 'SAM' population, two studies from India estimate the prevalence of severe anaemia in the child 'SAM' population between 52-67.3%, with a microcytic predominance (*Thakur et al., 2014*; *Arya et al., 2017*). Most of an infant's iron stores are endowed by the time of birth at term gestation with little derived from breast milk (*Ziegler, Nelson & Jeter, 2011*). This puts LBW infants at increased risk of iron deficiency, even if exclusively breastfed (*Mills & Davies, 2012*).

Our search identified eight MAs and five RCTs in pre-terms, infants and children but no articles specifically in malnourished populations. Significant heterogeneity was noted in terms of dosing schedule and outcomes assessed. Dose ranges were 2-4mg/kg in LBW, 7.5–10 mg/day in infants U6M and from <12.6–150 mg/day in children, although differing iron formulations were used with differing bioavailability of elemental iron. The most common dosing was 2mg/kg.

*Efficacy*

In children, of the two included MAs reporting on mortality, both generated insufficient data on which to estimate mortality (*Neuberger et al., 2016*; *De-Regil et al., 2011*). All four MAs in both pre-terms and children showed iron supplementation to be associated with an increase in mean haemoglobin concentration (GRADE High or Moderate where reported) (*Mills & Davies, 2012*; *De-Regil et al., 2011*; *Thompson, Biggs & Pasricha, 2013*; *Low et al., 2013*). Amongst infants, one RCT in Sweden and Honduras reported no impact on weight gain, but in sub-group analyses reported reduced length gain in Swedish infants 4–9 months of age and in Honduran non-anaemic infants (*Dewey et al., 2002*).

Psychomotor development in infants U6M was found to be significantly improved in the iron supplemented group in one MA (*Szajewska, Ruszczynski & Chmielewska, 2010*) but a further two RCTs of multiple micronutrients not included in this MA showed opposite effects in mean time to walking unassisted in subgroups analysing iron supplementation (*Olney et al., 2006*; *Katz et al., 2010*).

In children, three MAs showed inconsistent effects of iron on growth with one MA (wide age range from infants to >5 years) suggesting small benefits in terms of weight-for-age (WAZ) but reductions in some developed-setting subgroups in HAZ and rate of length gain (*Sachdev, Gera & Nestel, 2005*), one MA showing no effect on growth in children 2-5 years (GRADE - Very Low) (*Thompson, Biggs & Pasricha, 2013*), and one showing mild increases in HAZ and no impact on WAZ (5-12 years) (*Low et al., 2013*). There was evidence of improved cognitive performance in two MAs in those supplemented with iron from ages 2-12 years (GRADE –Very Low, where reported) (*Thompson, Biggs & Pasricha, 2013*; *Low et al., 2013*).

In LBW infants, the included MA (*Mills & Davies, 2012*) identified only one poor-quality RCT, out of 13 included studies reporting on growth, that showed improvements in growth (*Mills & Davies, 2012*) and a subsequent RCT has shown no benefit in terms of growth (*Berglund, Westrup & Domellöf, 2015*). This MA did not identify any studies comparing iron supplementation to placebo for neurodevelopmental outcomes and one study was identified that showed no difference in neurodevelopment between high- and low-dose iron but a higher incidence of an abnormal neurological examination at 5 years of age with late-onset supplementation of iron (*Mills & Davies, 2012*). Child behaviour but not intelligence scores were reported in one RCT as significantly better at seven-year follow-up of LBW infants supplemented with iron from birth compared with unsupplemented infants (*Berglund et al., 2018*).

### Safety

A MA of iron supplementation to children under 18 years in malaria-endemic areas showed no impact on overall malaria incidence and indeed 10% relative risk reduction of severe malaria was noted with iron supplementation (GRADE –High) (*Neuberger et al., 2016*). Iron was well tolerated in 10 of the 11 studies that documented adverse effects with only one MA noting an 11% increased risk of diarrhoea in iron supplemented groups ($p = 0.04$) (*Gera et al., 2002*).

### Summary

There is a lack of any strong evidence for benefits of iron supplementation in terms of mortality and morbidity but evidence of increased haemoglobin status and some neurodevelopmental benefits across age groups. Concerns, such as those raised in the WHO guidelines on iron supplementation in children (*World Health Organization, 2016*), exist about potential negative impacts of iron supplementation on growth, infection risk and malaria risk in malaria-endemic settings where regions where malaria prevention and treatment systems are not in place and/or where children are already iron replete (*Lönnerdal, 2017*). Routine use for all malnourished infants U6M cannot therefore be
recommended, but there can be exceptions for specific individuals and/or populations for treatment of iron deficiency.

Further trials are investigating potential alternatives to simple iron salts and ways to target iron therapy.

## Zinc

Zinc has received much attention for its role in the management of acute diarrhoea, but a plurality of roles in cell growth, immunity and metabolism continue to be identified (*Nissensohn et al., 2016*). Estimating prevalence of zinc deficiency is challenging for both logistical and assay availability reasons but recent estimates suggest 17.3% of the world's population is at high risk of deficiency with prevalence of inadequate zinc intake correlated to prevalence of childhood stunting (*Wessells & Brown, 2012*).

Our search identified a vast literature on zinc supplementation that proved challenging to summarise. We identified 24 relevant articles: two RCTs in infants and children with 'SAM' and four further RCTs in children over 6 months of age; one SR and seven RCTs in LBW infants; and two RCTs and nine MAs in non-malnourished infants and children. No articles were identified that looked at maternal post-natal zinc supplementation. In children, the most common zinc dosage was 20mg/day (range 5-40mg) with high dose usage in some RCTs up to 6mg/kg/day. In infants under 12 months, doses ranged from 1.78–20 mg/day and in LBW from 5–10 mg/day or 2mg/kg/day.

### *Efficacy*

Within the 'SAM' population, we identified two RCTs in infants U6M and children over 6 months of age showing a reduced total number of total infectious episodes (13 days fewer in the supplemented population over 90-days of follow-up, $p < 0.025$), reduction in number of diarrhoeal episodes (0.6 less than control group, $p = 0.04$) at 8 weeks follow-up following admission for diarrhoea, and evidence of slight length gain (4.4 mm greater compared to control group at 8 weeks follow-up, reported as $p < 0.05$) but not weight gain (*Castillo-Duran et al., 1987*; *Roy et al., 1999*). In children over the age of 6 months with 'SAM', two further RCTs identified significant increased anthropometric indices associated with zinc supplementation (one in length, two in weight and one in MUAC) (*Makonnen, Venter & Joubert, 2003*; *Umeta et al., 2000*). In one of these trials, examining 10mg zinc daily from admission to hospital until 90 days post-discharge, a significant 11% reduction in in-hospital mortality in zinc-supplemented children was also reported (*Makonnen, Venter & Joubert, 2003*) and in the other, there was reduced number of morbid episodes (cough, diarrhoea, fever, vomiting) in stunted children (*Umeta et al., 2000*). One further RCT showed no significant anthropometric impacts of high-dose compared to low-dose zinc (*Doherty et al., 1998*).

In a MA of the infant U6M population as a whole, no impact on all-cause mortality or diarrhoea duration or presence at day 7 was demonstrated (GRADE –Very Low and Low respectively) (*Lazzerini & Wanzira, 2016*). In combination with the child population a 13% reduction in pneumonia incidence (GRADE –Low) (*Lassi, Moin & Bhutta, 2016*) and slightly improved WAZ and WLZ (pooled effects +0.06 and +0.05 respectively) but not

MUAC were found (*Nissensohn et al., 2016*). A single RCT showed that zinc when used as an adjunct to treatment for neonatal sepsis resulted in significantly reduced mortality and improved mental development at 12 months (*Banupriya et al., 2018*).

Among children aged over 6 months, one MA showed an 18% reduction in all-cause mortality in those over 12 months (*Brown et al., 2009*), but two other MAs demonstrated no effect on mortality (GRADE –High and Very Low) (*Lazzerini & Wanzira, 2016*; *Mayo-Wilson et al., 2014*). Furthermore, three MAs demonstrated a reduction in incidence and duration of all-cause diarrhoea (GRADE –Moderate) (*Lazzerini & Wanzira, 2016*; *Brown et al., 2009*; *Mayo-Wilson et al., 2014*), and one MA identified a significant reduction in mortality when used as an adjunct to treatment for severe pneumonia (*Wang & Song, 2017*). A further four MAs noted improved, albeit across different anthropometric indices, with zinc supplementation (*Brown et al., 2009*; *Ramakrishnan, Nguyen & Martorell, 2008*; *Imdad et al., 2011*; *Liu et al., 2018*), with more marked improvements when supplemented after two years of age (*Liu et al., 2018*). No impacts on mental or psychomotor development were demonstrated by one MA (GRADE –Moderate) (*Gogia & Sachdev, 2012*).

In the LBW population, concerning indices of growth, one SR of three trials identified no impact of zinc supplementation on length or weight (*Gulani, Bhatnagar & Sachdev, 2011*), but four of five RCTs identified by our search demonstrate significant effects in terms of length and weight gain (*El Sayed & Elghorab, 2016*; *El-Farghali et al., 2015*; *Terrin et al., 2013*; *Friel et al., 1993*; *Díaz-Gómez et al., 2003*). The same SR included one trial that showed no benefit in terms of mortality, whereas our search identified two RCTs suggesting a reduction in mortality with zinc supplementation by 58–68% (*Terrin et al., 2013*; *Sazawal et al., 2001*). Concerning morbidity outcomes, one SR identified two trials showing no overall impacts on number of diarrhoeal illnesses but fewer days of diarrhoea following cessation of breastfeeding and one trial showing no impact on acute lower respiratory tract infection incidence (*Gulani, Bhatnagar & Sachdev, 2011*). Furthermore, one RCT showed a reduction in combined neonatal morbidities (late-onset sepsis, necrotising enterocolitis, bronchopulmonary dysplasia etc.) with zinc supplementation (*Terrin et al., 2013*). Finally, one RCT reporting on neurodevelopmental outcomes showed increased alertness and attention at term corrected gestational age and reduced hyper-excitability at three months follow-up in zinc supplemented infants (*Mathur & Agarwal, 2015*).

### Safety

Adverse effects and safety concerns associated with zinc supplementation were poorly reported with only 10 of 24 included articles explicitly mentioning this. Of articles that documented adverse effects or none, one RCT in Bangladesh, of 141 children 6 months–3 years old with 'SAM' identified a 4.5 times increased mortality with high-dose (6.0mg/kg for 15+ days) compared to low-dose (1.5 mg/kg for 15 days) zinc supplementation ($p = 0.03$) (*Doherty et al., 1998*) and two MAs in infants and children identified a significant 29–57% increased risk of one or more episodes of vomiting upon commencement of zinc supplements (GRADE –High and Moderate) (*Lazzerini & Wanzira, 2016*; *Mayo-Wilson et al., 2014*).

*Summary*

Consistent evidence across age groups exists of zinc supplementation being associated with reduced morbidity and improved anthropometry, whilst mortality and neurodevelopmental impacts are more unclear. Zinc should not be supplemented as a high-dose, given mortality concerns, and its tolerability should be considered in the context of vomiting risk. Zinc should be supplemented as per diarrhoea guidelines for all severely malnourished infants U6M with diarrhoeal illnesses (*World Health Organization, 2005*). In cases where severely malnourished infants U6M are not affected by diarrhoea, in the absence of further evidence, we suggest that 2mg/day (1 recommended daily allowance (*Institute of Medicine , US*)) of zinc be supplemented with other micronutrients in regions where zinc deficiency has been documented. This is because breastmilk zinc concentrations have been shown to be insufficient in zinc deficient mothers (*Dumrongwongsiri et al., 2015*; *Yalcin, Yalcin & Gucus, 2015*).

Research is urgently needed to establish zinc requirements for malnourished infants U6M and the fact that there is some evidence of mortality benefit when supplemented over an extended period suggests that randomised trials are warranted in this age group.

## Folic acid

Folic acid is essential for DNA synthesis and repair, erythropoiesis and cellular metabolism and deficiency is clinically associated with megaloblastic anaemias and foetal neural tube defects (*Bailey, West & Black, 2015*). Characterising the burden of folic acid deficiency in children is challenging given differing definitions, but, amongst pre-school children in sub-Saharan Africa, estimates range from 0–8.5% with the exception of one study in Gambia which estimated a 24% prevalence (*Kupka, 2015*).

Folic acid has garnered much attention regarding pre-conceptual and pre-natal supplementation, but our search identified comparatively little with regard to child or post-natal supplementation: no MAs in infants and children and 10 RCTs across pre-terms, infants and both well- and malnourished children. Folic acid supplementation varied from 50–250 µg/day among trials in LBW and between 50-150 µg/day in those among children.

*Efficacy*

Folic acid supplementation was shown in two RCTs in infants and children to have no impact on mortality (*Sazawal et al., 2006*; *Tielsch et al., 2006*). In children over 6 months of age, two RCTs showed reduced incidence of acute diarrhoeal disease and lower respiratory tract infection (*Tielsch et al., 2006*; *Taneja et al., 2013*) and one RCT showed increased total weight gain and WAZ scores with folic acid supplementation compared to placebo (*Medeiros et al., 2015*).

Amongst LBW infants, of three RCTs included all showed no impact of folic acid on weight gain, one noticed length gains in a subgroup of infants of birthweight >1750g, and one showed no improvement in infectious disease incidence or haemoglobin status (*Foged et al., 1989*; *Kendall et al., 1974*; *Ek et al., 1984*).

*Safety*

A safety debate around folic acid supplementation has centred on its usage in malaria-endemic settings as malaria parasites can utilise exogenous folate during co-administration with sulfadoxine/pyrimethamine (SP) (a folate antagonist used in the prevention and treatment of malaria). In a paired set of RCTs studying folic acid supplementation amongst children 1-36 months, the Tanzanian trial was stopped early because of increased adverse events and hospital admissions in the supplemented arm (*Sazawal et al., 2006*) but this was not replicated in the Nepalese study (*Tielsch et al., 2006*). Subsequent post-hoc analysis of the Tanzanian study showed adverse events were not associated with SP co-administration. Three further RCTs looking at SP and folic acid co-administration showed some evidence of parasitological but not clinical treatment failure with SP and folic acid co-administration (*Carter et al., 2005*; *Van Hensbroek et al., 1995*; *Mulenga et al., 2006*).

Folic acid was well tolerated in other studies with the exception of one RCT amongst Indian children that showed a two times increased incidence of persistent diarrhoea in the supplemented group (*Taneja et al., 2013*), something that was not replicated in the two other RCTs reporting on diarrhoea incidence.

*Summary*

There is limited evidence of benefit in the child population in terms of morbidity and growth and therefore this is not recommended as a routine intervention for malnourished infants U6M. Any benefits that may exist are not consistent across age groups. Safety in a malaria endemic setting remains uncertain but is less concerning given SP is no longer routinely used for the treatment of malaria. Safety of folic acid supplementation should be considered in areas where SP is used for Seasonal Malarial Chemoprophylaxis (SMC) or Intermittent Preventive Treatment in pregnancy (IPTp).

Further studies investigating the role of folic acid in malnourished infants U6M are warranted but the evidence presented does not identify this as a priority area for research.

## Maternal supplementation of macronutrients or multiple micronutrients

Given the nutritive demands on the breastfeeding mother, maternal micronutrient and macronutrient supplementation during breastfeeding is seen as a potentially promising approach to improving both maternal and infant nutritional status and multiple micronutrient and macronutrient supplementation has received increasing attention in recent years.

Our search identified four MAs and one RCT covering the topics of post-natal multiple micronutrient and polyunsaturated fatty acid (PUFA) supplementation. Articles pertaining to maternal supplementation with single micronutrients are included in the section relevant to that micronutrient. Within these included studies the post-natal period was inconsistently defined but supplementation typically occurred within 2 weeks of delivery and up to 4 months post-partum.

*Efficacy*

Maternal post-natal multiple micronutrient supplementation was associated with no quantitative evidence of improvement in infant and child mortality or morbidity outcomes (*Abe et al., 2016*). One of the three MAs looked at PUFA supplementation compared to placebo to breastfeeding mothers postnatally and demonstrated no benefit in terms of infant length, weight and head circumference (GRADE Moderate) (*Delgado-Noguera et al., 2015*). These results of no significant growth effects were supported in the two other MAs but the population included supplementation during both gestation and lactation and no postnatal subgroup analysis was presented (*Quin et al., 2016*; *Li et al., 2018*). A statistically significant improvement was noted in indices of child attention beyond 24 months of age in the first of these MAs (GRADE Low) (*Delgado-Noguera et al., 2015*). An RCT of maternal calorie supplementation in conjunction with a breastfeeding support intervention showed no improvement in WAZ or LAZ but increased infant breast milk intake and a two times increase in EBF (likely secondary to the support intervention) was demonstrated (*Huynh et al., 2017a*).

*Safety*

No clinically detectable adverse effects were reported in any of the included studies.

*Summary*

We found insufficient evidence of the benefits of maternal supplementation to infants to justify routine use in current programming for malnourished infants U6M. However, potential benefits to the mother, not included in this review, should be considered and evaluated in more detail in further research to inform decisions in this area.

## Oral Rehydration in severe malnutrition complicated by diarrhoea

Diarrhoea affects a large proportion of children with severe malnutrition admitted to inpatient care. Studies in sub-Saharan Africa identify the prevalence of diarrhoea at admission to hospital with 'SAM' at 49-67.3%, with its presence being associated with increased inpatient mortality rates (19% vs 9%, OR 2.5) (*Talbert et al., 2012*; *Irena, Mwambazi & Mulenga, 2011*). The current WHO guidelines for the management of diarrhoea in 'SAM' in children over 6 months recommend oral rehydration rather than intravenous rehydration, unless shock exists, due to the theoretical concerns of causing fluid overload and precipitating heart failure (*World Health Organization, 2013b*). ReSoMal is the recommended form of ORS in 'SAM' complicated by diarrhoea, except in cases of cholera. It differs from the standard WHO hypo-osmolar ORS in having a lower sodium, higher potassium and higher glucose concentrations as well as a higher osmolality. The comparative compositions are presented in Table 4. The rationale for these differing compositions being that severely malnourished children would be predisposed to fluid retention due to their already high intracellular sodium concentrations, again risking fluid overload and heart failure (*World Health Organization, 2013a*), although the evidence underlying this is contentious (*Houston et al., 2017*).

Infants in the first few months of life are at increased risk of dehydration during diarrhoea episodes because of their relatively higher body surface area and difficulty

**Table 4** Compositions of commonly used oral rehydration solutions (ORS) (*World Health Organization, 2013a*).

|  | WHO Standard ORS | ReSoMal |
|---|---|---|
| **Osmolarity (mOsm/l)** | 245 | 300 |
| **Sodium (mmol/l)** | 75 | 45 |
| **Potassium (mmol/l)** | 20 | 40 |
| **Chloride (mmol/l)** | 65 | 76 |
| **Glucose (mmol/l)** | 75 | 125 |

drinking enough to match losses. They are also more susceptible to water and sodium retention because of immature renal sodium and water excretion mechanisms both in the kidney and in hormonal control (*Murtaza et al., 1987*; *Marin et al., 1987*; *Elliott et al., 1989*). Theoretically, this may reduce the risk of hyponatraemia during rehydration with lower osmolarity ORS compared with older children. The 2013 WHO SAM guidelines update makes no comment on how the rehydration of infants U6M with diarrhoea should be managed. However, there is explicit mention, albeit concerning their nutritional management, that infants *'should not be given undiluted F-100 at any time (owing to the high renal solute load and risk of hypernatraemic dehydration)'* (*World Health Organization, 2013a*); a factor that should be considered when thinking about oral rehydration in this age group. There appears to be widespread consensus that standard WHO ORS should be used in the rehydration of infants U6M without 'SAM' and that breastfeeding should be continued throughout the episode of acute gastroenteritis (*World Health Organization, 2013a*; *Gregorio, Dans & Silvestre, 2012*; *Shane et al., 2017*; *Guarino et al., 2018*).

Our search focused on the evidence for the usage of ReSoMal for oral rehydration in severe malnutrition complicated by diarrhoea. Comparisons of other formulations of ORS, polymer-based ORS and intravenous rehydration strategies were considered beyond the scope of this review.

We identified one recent SR (*Huynh et al., 2017a*) that included two RCTs in the Asian inpatient setting comparing ReSoMal to WHO ORS in children over 6 months, which were assessed to be of low risk of bias (*Alam et al., 2003*; *Kumar et al., 2015*). No articles looking at ReSoMal in infants U6M were identified.

### Efficacy

Both studies reported on time to rehydration, with *Alam et al. (2003)* showing no difference between ReSoMal and ORS and *Kumar et al. (2015)* reporting a shorter time to rehydration with ReSoMal (16.1 h vs 19.6 h $p = 0.036$). Stool frequency and number of patients requiring IV fluids after attempting oral rehydration was shown to be similar between treatment groups in both studies. Only *Alam et al. (2003)* reported on mortality, reporting no deaths. Neither study reported on anthropometric or neurodevelopmental outcomes.

### Safety

The primary outcome of the included SR was the occurrence of hyponatraemia. *Alam et al. (2003)* ReSoMal was associated with a higher incidence of severe hyponatraemia (Na<120mmol compared to an older WHO ORS formulation (5% versus 2%,) a lower

mean serum sodium at 24 and 48 h ($p < 0.01$ and $<0.001$ respectively) and with one episode of hyponatraemic seizures ($n = 130$). Similarly, *Kumar et al. (2015)*, comparing ReSoMal with standard ORS, showed ReSoMal to be associated with increased incidence of hyponatraemia (15.4% vs 1.9%, $n = 110$). *Alam et al. (2003)* reported there to be no difference in frequency of fluid overload between groups.

### *Summary*

There are no studies of ReSoMal in infants U6M. Limited evidence from inpatient studies of older children with 'SAM' suggests that ReSoMal is of similar efficacy in terms of rehydration to standard ORS but that there are significant safety concerns in terms of risk of hyponatraemia. On the basis of current evidence, and the fact that infants in the first few months of life are at increased risk of water and salt retention due to immature hormonal and renal excretion mechanisms, there is no reason to change current recommendations for use of ReSoMal in malnourished infants U6M. This age group may differ from older children in both risks and responses to treatment and is thus a priority area for clinical trials.

## CONCLUSIONS

This review has collated, reviewed and summarised the evidence-base for a selection of common medical interventions (antibiotics, de-worming, infant micronutrient supplementation, maternal macro- and micro-supplementation and ReSoMal) that may be considered for use in infants U6M identified by current screening criteria as severely malnourished. A key finding is the lack of direct evidence for this population group. In its absence, we have identified evidence in closely-related populations, exploring consistency of effect in terms of both efficacy and safety. Even for these groups, including LBW infants and malnourished children over the age of 6 months, the evidence base is neither strong nor extensive. It does however allow us to make some tentative recommendations and establishes an initial evidence base from which policy discussions about best approaches to managing infants U6M with severe malnutrition can begin.

The scope of this review is deliberately broad, particularly in using a non-specific definition of severe malnutrition and broad age group categories to reflect current WHO subdivisions. The utility of this review is derived from it gathering a vast body of evidence of disparate interventions, populations and study outcomes which may help frame policy and programming discussions as well as future research. Given such breadth, it was not possible to perform a full systematic search strategy and MA for each intervention nor was it possible to do a comprehensive assessment of quality of evidence, something that would be warranted in the future. Guidance for programmers is urgently needed, and whilst we have presented tentative recommendations, the weak and disparate evidence-base would benefit from wider expert consultation and consensus beyond those directly engaged in this review.

It is notable that of included articles, most showed no effect of the investigated intervention and amongst those that did, the effects were small with broad confidence intervals. There is a lack of accurate global micronutrient profiling, partly due to technical

difficulties in measurement and capacity for assessment in low-resource settings, and as a result, the majority of studies look at the impact of mass supplementation without knowledge of whether it is a nutrient deficient or replete population. Studies may therefore be underpowered to detect benefits at a population level and fail to detect important effects among specifically micronutrient deficient infants and children. In addition, on reviewing the doses and dosing regimens of included studies, there is significant heterogeneity in dosing between trials within age groups, let alone across age groups. This makes creating accurate estimates of effect challenging and leaves policy makers with difficulty as to how to interpret these null or marginal effects, and programmers uncertain as to what dosage to implement in the context of their specific population or sub-population.

Little is known about micronutrient status and needs of infants U6M identified as malnourished. Although it is likely that they fall somewhere between those of LBW infants and malnourished children over 6 months of age, they may vary between those born LBW or not, or in relation to maternal capacity to develop foetal pre-natal stores, and by exclusivity of breastfeeding. Given this, we have presented the evidence for medical interventions in these two adjacent populations groups; the implications of these findings on the population of malnourished U6M and how this should shape future programming remains to be determined.

The review raises several key research and policy questions which warrant urgent discussion and evaluation including: what is the micronutrient status of infants U6M with severe malnutrition and what interventions might be effective in their recovery, what is the optimal oral rehydration protocol for this age group, how might medical interventions practically be delivered alongside supporting/re-establishing EBF and finally what interventions can safely form interim policy in the absence of direct evidence?

## ACKNOWLEDGEMENTS

The authors also thank WHO (Department of Maternal, Newborn, Child and Adolescent Health and the Department of Nutrition for Health and Development) and the Gates Foundation for commissioning an update to this review, which was presented at a January 2019 technical consultation on 'Research priorities to prevent, identify and manage young infants with growth failure in the first 6 months of life'

### Funding

Timothy Campion-Smith and Marie McGrath were supported by Irish Aid funding to Emergency Nutrition Network to undertake this review. James Berkley is supported by the Bill & Melinda Gates Foundation (grant number OPP1131320) to the Childhood Acute Illness & Nutrition (CHAIN) Network and the MRC/DfID/Wellcome Trust Global Health Trials Scheme (grant number MR/M007367/1). The WHO (Department of Maternal, Newborn, Child and Adolescent Health and the Department of Nutrition for Health and Development) and the Gates Foundation provided funding to all authors to update this

review. The funders had no role in study design, data collection and analysis, decision to publish, or preparation of the manuscript.

### Grant Disclosures

The following grant information was disclosed by the authors:
Emergency Nutrition Network.
Bill & Melinda Gates Foundation (grant number ): OPP1131320.
Childhood Acute Illness & Nutrition (CHAIN) Network and the MRC/DfID/Wellcome Trust Global Health Trials Scheme: MR/M007367/1.
The WHO (Department of Maternal, Newborn, Child and Adolescent Health and the Department of Nutrition for Health and Development).
Gates Foundation.

### Competing Interests

Marie McGrath is employed by the Emergency Nutrition Network.

Timothy Campion-Smith is employed by Oxford University Hospitals and was employed by Emergency Nutrition Network as a consultant to undertake this review.

### Author Contributions

- Timothy J Campion-Smith conceived and designed the experiments, performed the experiments, analyzed the data, prepared figures and/or tables, authored or reviewed drafts of the paper, and approved the final draft.
- Marko Kerac, Marie McGrath and James A. Berkley conceived and designed the experiments, authored or reviewed drafts of the paper, and approved the final draft.

### Data Availability

This is a literature review article and did not generate raw data.

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
