# Peer review of "Antimicrobial and micronutrient interventions for the management of infants under 6 months of age identified with severe malnutrition: a literature review"

_PeerJ, doi:10.7717/peerj.9175_

## Round 0.1 · original submission · Major Revisions

Dear Dr. Campion-Smith,

Your manuscript entitled " Medical interventions for the management of infants under 6 months of age identified with severe malnutrition: a literature review" which you submitted to PeerJ, has been reviewed by the editor and 3 expert reviewers in the field.

I regret to inform you that the Reviewers have raised significant concerns that need to be addressed before the manuscript can be considered further. Since the reviewers found your topic interesting and timely, I would be willing to reconsider if you wish to undertake major revisions and resubmit.

If you decide to resubmit the revised version, please summarize all the improvements made in the new version and give answers to all critical points raised in the reviewers’ report in an accompanying letter. Please copy and paste each and every reviewer's comment above your response. If you feel any of their points are inappropriate, you are certainly free to provide rebuttal in your covering letter.

I would strongly encourage you to consider re-conceptualizing your literature study, identifying a clear objective/focus of the review and being more consistent with the terminology used when referring to malnutrition and ages/stages of children. All these changes are necessary to improve the accessibility of the information collected in the review article.

Please note that resubmitting your manuscript does not guarantee eventual acceptance. Since the requested changes are major, the revised manuscript will undergo a second round of review by the same reviewers. I must emphasize that the acceptability of the revision will depend upon the resolution of the points raised by the reviewers.

Sincerely yours,

Stefano Menini

Reviewer 1 ·

Basic reporting

I have put everything in one doc and pasted under each; I had no time to recategorize it in your format.
The review is dealing with a very interesting topic that needs more attention and is therefore very timely.
This review attempts to do a lot in one review: It might want to do too much and therefore the reader gets easily lost. The title is promising but possibly misleading. It is more about interventions related to impact on nutritional status than medical interventions as such.

Though it is a very courageous attempt to do this literature study it would benefit from perhaps less ambitious goals. The authors opt to look for interventions and what is known in adjacent age groups in relation to ‘severe malnutrition’. It is not fully clear what the authors anticipated as the terminology varies in this document. The authors use sometimes ‘severe malnutrition’ but it is unclear whether they refer to severe acute malnutrition or any type of severe malnutrition. It becomes more confusing when terms like
‘Acute wasting, severe malnutrition, acutely malnourished, SAM’ (the first and third include moderate acute malnutrition) are used. The authors seem occasionally to use this mix of terms interchangeably.

The authors use various age groups/terms (infant, child) without precisely defining these. What is the definition of an infant in this context: 0-12 months? 0-6 months? What is the definition in this context of a “child”? Do they mean anyone 6 months to 18 years? or 6-59 months? In line 163 it is stated children of any age. This leads to confusion.

Though this is not a ‘must’ but the authors could consider not to take the intervention as a starting point but the physiological group: and list all what is known for that group related to malnutrition: e.g. LBW through pre-term; LBW through SGA; 0-6 months (malnourished and non malnourished – ideally more specific: SAM and/or MAM); 6-59 months (malnourished and non malnourished – ideally more specific: SAM and/or MAM). It is unclear why the authors included information on any child older than 59 months. For example: references are made to 5-12 years or 2-12 years but the purpose is to look at adjacent age groups (e.g. line 440-441). It is questionable to consider a child 5 years and older as an adjacent group.

The authors describe the efficacy of various interventions (line 120) but as the efficacy is related to a desired outcome it is not always clear what outcome is referred to in the different sections.
It would have been helpful to have the results marked in a table and not just a summary table:
e.g. a table with column headings such as:
subject/age group/biological group, intervention, sort of study, objective intervention, description of subjects (SAM, MAM, stunted, etc), outcomes, etc
The authors do not discuss any neonatal period and put everything <6 months in one group. The neonatal period is a distinctive period and should be acknowledged as such. Also, the nutritional needs in that group in infants that are preterm are different from other neonates.
It is unclear why the authors decided to exclude the impact of breastmilk fortifiers.

The paper reads more like a report from a lit review than a manuscript for a peer reviewed journal. Sometimes it reads like a long list of citations from the reviewed journals causing lack of uniformity and cohesion. The introduction is not very clear and many passages raise questions on what the authors exactly meant to say. It would benefit to summarize what is known about the 0-6 months and SAM (or both SAM and MAM) in a table.
This manuscript needs thorough editing and text can be written more concise. There are various paragraphs where words are missing.
Additionally, the manuscript would benefit from some clarifications/ corrections:
The dosages of various interventions are not always clear daily, single intake/bolus e.g. lines 359-361. The references need some additional reviewing by the authors: here and there missing pages, missing journal (e.g. references 4, 8, 16, 19, etc).
Line 422: <12.6-150mg/day in children: 150 mg per day???
Table 1 : do the authors mean >6 months until 18 years, or 6-59 months?
The titles of the tables/figures are not always clear and would benefit with some more precise description.

Though a good attempt to provide more clarity on what is known/needed for the 0-6 months old malnourished children, this review needs more work to make the information it found more accessible.

Experimental design

The review is dealing with a very interesting topic that needs more attention and is therefore very timely.
This review attempts to do a lot in one review: It might want to do too much and therefore the reader gets easily lost. The title is promising but possibly misleading. It is more about interventions related to impact on nutritional status than medical interventions as such.

Though it is a very courageous attempt to do this literature study it would benefit from perhaps less ambitious goals. The authors opt to look for interventions and what is known in adjacent age groups in relation to ‘severe malnutrition’. It is not fully clear what the authors anticipated as the terminology varies in this document. The authors use sometimes ‘severe malnutrition’ but it is unclear whether they refer to severe acute malnutrition or any type of severe malnutrition. It becomes more confusing when terms like
‘Acute wasting, severe malnutrition, acutely malnourished, SAM’ (the first and third include moderate acute malnutrition) are used. The authors seem occasionally to use this mix of terms interchangeably.

The authors use various age groups/terms (infant, child) without precisely defining these. What is the definition of an infant in this context: 0-12 months? 0-6 months? What is the definition in this context of a “child”? Do they mean anyone 6 months to 18 years? or 6-59 months? In line 163 it is stated children of any age. This leads to confusion.

Though this is not a ‘must’ but the authors could consider not to take the intervention as a starting point but the physiological group: and list all what is known for that group related to malnutrition: e.g. LBW through pre-term; LBW through SGA; 0-6 months (malnourished and non malnourished – ideally more specific: SAM and/or MAM); 6-59 months (malnourished and non malnourished – ideally more specific: SAM and/or MAM). It is unclear why the authors included information on any child older than 59 months. For example: references are made to 5-12 years or 2-12 years but the purpose is to look at adjacent age groups (e.g. line 440-441). It is questionable to consider a child 5 years and older as an adjacent group.

The authors describe the efficacy of various interventions (line 120) but as the efficacy is related to a desired outcome it is not always clear what outcome is referred to in the different sections.
It would have been helpful to have the results marked in a table and not just a summary table:
e.g. a table with column headings such as:
subject/age group/biological group, intervention, sort of study, objective intervention, description of subjects (SAM, MAM, stunted, etc), outcomes, etc
The authors do not discuss any neonatal period and put everything <6 months in one group. The neonatal period is a distinctive period and should be acknowledged as such. Also, the nutritional needs in that group in infants that are preterm are different from other neonates.
It is unclear why the authors decided to exclude the impact of breastmilk fortifiers.

The paper reads more like a report from a lit review than a manuscript for a peer reviewed journal. Sometimes it reads like a long list of citations from the reviewed journals causing lack of uniformity and cohesion. The introduction is not very clear and many passages raise questions on what the authors exactly meant to say. It would benefit to summarize what is known about the 0-6 months and SAM (or both SAM and MAM) in a table.
This manuscript needs thorough editing and text can be written more concise. There are various paragraphs where words are missing.
Additionally, the manuscript would benefit from some clarifications/ corrections:
The dosages of various interventions are not always clear daily, single intake/bolus e.g. lines 359-361. The references need some additional reviewing by the authors: here and there missing pages, missing journal (e.g. references 4, 8, 16, 19, etc).
Line 422: <12.6-150mg/day in children: 150 mg per day???
Table 1 : do the authors mean >6 months until 18 years, or 6-59 months?
The titles of the tables/figures are not always clear and would benefit with some more precise description.

Though a good attempt to provide more clarity on what is known/needed for the 0-6 months old malnourished children, this review needs more work to make the information it found more accessible.

Validity of the findings

The review is dealing with a very interesting topic that needs more attention and is therefore very timely.
This review attempts to do a lot in one review: It might want to do too much and therefore the reader gets easily lost. The title is promising but possibly misleading. It is more about interventions related to impact on nutritional status than medical interventions as such.

Though it is a very courageous attempt to do this literature study it would benefit from perhaps less ambitious goals. The authors opt to look for interventions and what is known in adjacent age groups in relation to ‘severe malnutrition’. It is not fully clear what the authors anticipated as the terminology varies in this document. The authors use sometimes ‘severe malnutrition’ but it is unclear whether they refer to severe acute malnutrition or any type of severe malnutrition. It becomes more confusing when terms like
‘Acute wasting, severe malnutrition, acutely malnourished, SAM’ (the first and third include moderate acute malnutrition) are used. The authors seem occasionally to use this mix of terms interchangeably.

The authors use various age groups/terms (infant, child) without precisely defining these. What is the definition of an infant in this context: 0-12 months? 0-6 months? What is the definition in this context of a “child”? Do they mean anyone 6 months to 18 years? or 6-59 months? In line 163 it is stated children of any age. This leads to confusion.

Though this is not a ‘must’ but the authors could consider not to take the intervention as a starting point but the physiological group: and list all what is known for that group related to malnutrition: e.g. LBW through pre-term; LBW through SGA; 0-6 months (malnourished and non malnourished – ideally more specific: SAM and/or MAM); 6-59 months (malnourished and non malnourished – ideally more specific: SAM and/or MAM). It is unclear why the authors included information on any child older than 59 months. For example: references are made to 5-12 years or 2-12 years but the purpose is to look at adjacent age groups (e.g. line 440-441). It is questionable to consider a child 5 years and older as an adjacent group.

The authors describe the efficacy of various interventions (line 120) but as the efficacy is related to a desired outcome it is not always clear what outcome is referred to in the different sections.
It would have been helpful to have the results marked in a table and not just a summary table:
e.g. a table with column headings such as:
subject/age group/biological group, intervention, sort of study, objective intervention, description of subjects (SAM, MAM, stunted, etc), outcomes, etc
The authors do not discuss any neonatal period and put everything <6 months in one group. The neonatal period is a distinctive period and should be acknowledged as such. Also, the nutritional needs in that group in infants that are preterm are different from other neonates.
It is unclear why the authors decided to exclude the impact of breastmilk fortifiers.

The paper reads more like a report from a lit review than a manuscript for a peer reviewed journal. Sometimes it reads like a long list of citations from the reviewed journals causing lack of uniformity and cohesion. The introduction is not very clear and many passages raise questions on what the authors exactly meant to say. It would benefit to summarize what is known about the 0-6 months and SAM (or both SAM and MAM) in a table.
This manuscript needs thorough editing and text can be written more concise. There are various paragraphs where words are missing.
Additionally, the manuscript would benefit from some clarifications/ corrections:
The dosages of various interventions are not always clear daily, single intake/bolus e.g. lines 359-361. The references need some additional reviewing by the authors: here and there missing pages, missing journal (e.g. references 4, 8, 16, 19, etc).
Line 422: <12.6-150mg/day in children: 150 mg per day???
Table 1 : do the authors mean >6 months until 18 years, or 6-59 months?
The titles of the tables/figures are not always clear and would benefit with some more precise description.

Though a good attempt to provide more clarity on what is known/needed for the 0-6 months old malnourished children, this review needs more work to make the information it found more accessible.

Reviewer 2 ·

Basic reporting

General comments
The manuscript in general well written and well organized, the language also good.
Abstract
Line 1- 3, the title said the study groups are under 6months infants but on the result part (line 45) said “Ninety-four articles were identified for inclusion within this review, none of which
Studied interventions solely in acutely malnourished infants U6M”, so how the studies were included for review since they weren’t done on U6months?

Experimental design

Survey methodology
In general, the method part lacks a clear description of the objective of the review.
Line 131-132, infants U6M and children with and without malnutrition included. Why without malnutrition children and U6M included?
Line 138-139, anthropometric changes what are these anthropometric changes? Neurodevelopment = how assessed?
Line 145 149, is not clear and quite hard to understand.
“Where an RCT was included in a subsequent MA, the article was not included unless it contained a subgroup of interest (e.g. malnourished infants and children) that was not the primary population of interest in the MA”
“Similarly, where a study looked at multiple micronutrient interventions, this was
149 included if subgroups were presented analyzing the effect of a single micronutrient.”

Validity of the findings

Result
In general, the result part bulky and can’t relate the review result with the title of the review. As the manuscript described that the review was done on U6M severely malnutrition groups but the reviewed MA, SR, RCT was not in these target groups. Thus, it’s difficult to comment on the result part.

·

Basic reporting

For infants under 6 months whole management varies whether there is prospects of BF or there is no prospects.
Whole review does not talk about this differentiation

Experimental design

Appropriate

Validity of the findings

Applicable

Additional comments

WHO 2013 updates has given several additional criteria for inpatient care . There is a need to have recommendations for each condition

---

## Round 0.2 · accepted · Accept

Dear Dr. Campion-Smith,

I am pleased to inform you that the revision of your manuscript entitled “Antimicrobial and micronutrient interventions for the management of infants under 6 months of age identified with severe malnutrition: a literature review " now makes it acceptable for publication in PeerJ. I appreciate very much your making the suggested revisions.

Yours sincerely,

Stefano Menini